# Facilitators and barriers for completion of the diagnostic process among people with presumed tuberculosis in Central Uganda

Rebecca Nuwematsiko[1]*, Lynn Atuyambe[2], Vicent Kasiita[3], Noah Kiwanuka[4], Angella Musiimenta[5], Elizeus Rutebemberwa[6], Esther Buregyeya[1]

1 Makerere University, School of Public Health, Department of Disease Control and Environmental Health, Kampala, Uganda, 2 Makerere University, School of Public Health, Department of Community Health and Behavioural Sciences, Kampala, Uganda, 3 Infectious Diseases Institute, Kampala, Uganda, 4 Makerere University, School of Public Health, Department of Epidemiology and Biostatistics, Kampala, Uganda, 5 Mbarara University of Science and Technology, Kampala, Uganda, 6 Department of Health Policy, School of Public Health, Planning and Management, Makerere University, Kampala, Uganda

* rnuwematsiko@musph.ac.ug

## Abstract

Uganda has improved its tuberculosis (TB) diagnostic processes over the years since the scale up of the Xpert MTB/RIF testing in 2012. However, there are continued delays in diagnosis and missing people with TB who are either not diagnosed or notified. We explored facilitators and barriers for completion of the TB diagnostic process among people with presumed TB in selected health facilities in Central Uganda. This was a qualitative exploration involving 25 in-depth interviews with people with presumed TB and six sex stratified focus group discussions with people with TB who had recently initiated treatment. We also conducted 20 key informant interviews with health workers providing TB services. All interviews and discussions were audio recorded and transcribed verbatim. Thematic analysis was carried out using Atlas. ti software version 6.0 guided by the constructs of the socio-ecological model. Key facilitators for completion of the TB diagnostic process included; individual factors (persistent symptoms and the desire to get better, obtaining same day results and prior TB knowledge); community (social support); and health system factors (caring health workers and calling of patients to collect results). Barriers were; individual factors (TB and HIV stigma, inability to produce sputum, lack of transport to return to the health facility); and health system factors (long turnaround time of results, stock out of supplies, unclear appointment for collection of results, inadequate patient contact details and negative health worker attitude). Completion of the TB diagnostic process is influenced by individual, health system and community related factors. To enhance completion, there is need for availing same day results, making clear appointments for collection of results and improving health worker attitudes at the health facility level. At the individual and community level, TB/HIV stigma reduction interventions,

**Data availability statement:** All the data and materials supporting this manuscript are within the paper and the supplementary file, also accessed at the Zenodo repository (https://doi.org/10.5281/zenodo.16944363). More information can be requested through the Research and Ethics Committee of Makerere University School of Public Health. They can be contacted at sphrecadmin@musph.ac.ug, +256-701-888-163.

**Funding:** This study was funded under the EDCTP2 programme supported by the European Union (grant number: TMA2018SF-2472-MILEAGE4TB). EB was the recipient of the grant. The funders had no role in study design, data collection and analysis, decision to publish, or preparation of the manuscript.

**Competing interests:** The authors have declared that no competing interests exist.

community health education on TB and provision of social support to patients with presumed TB should be emphasized.

## Introduction

Tuberculosis (TB) remains a significant global health challenge and in 2023 it was ranked the leading cause of death from a single infectious agent [1]. The Sustainable Development Goals (SDGs) call on all countries to find the missing TB patients in order to end the TB epidemic by 2035 [2,3]. This has however been challenged by the continued loss of people with presumed TB before they complete the diagnostic process (pre-diagnosis loss) and before initiating treatment if diagnosed with TB (pre-treatment loss) [4,5]. People with presumed TB are adults or children with symptoms suggestive of TB; cough, fever, night sweat, weight loss and previous contact with a TB patient. Pre-diagnosis and pre-treatment loss of patients facilitates continued spread of TB in the community, poor treatment outcomes for persons with TB and death [4–8]. In Sub-Saharan Africa, 48.5% of people with presumed TB are lost before receiving a diagnosis according to a systematic review and metanalysis of studies [9]. This burden of people with presumed TB getting lost before being diagnosed is shared across all TB-burden countries although it is highest in high-TB burden countries compared to low-burden countries [4,6].

Uganda is among the 30 TB/HIV high burden countries [1] with an incidence of 198 cases per 100,000 people [10]. In 2023, the country's TB case notification rate decreased to 178 cases per 100,000 people up from 200 cases per 100,000 in 2022, indicating a gap in diagnosis and notification [1,10]. Relatedly, studies in Uganda by Ekuka et al., [11] and Nuwematsiko et al., [12] showed that 40.7% and 28% of patients who are presumed with TB do not complete the diagnostic process respectively. In order to improve case detection and notification, the country adopted the World Health Organization (WHO) TB systematic screening and testing strategy at all health facilities but its implementation is low at 75.1%, below the 100% target [10,13]. Although this intervention is in place to foster increased case detection, people with presumed TB continue to be lost in Uganda before completing the diagnostic process and initiating treatment. Completion of the TB diagnostic process refers to undertaking the requested TB investigations and obtaining results from the test within 30 days from the date of being identified with presumed TB [12].

There is paucity of data on factors for the loss of patients before completing the TB diagnostic process in Uganda. A previous study by Zawedde-Muyanja et al., [14] explored factors for the loss of patients largely before treatment initiation leaving gaps on an in-depth analysis of the phase before one is tested and diagnosed for TB. Understanding factors that influence completion of the TB diagnostic process will provide in-depth contextual insights into these factors which will inform local strategies for improving the Uganda TB response thus contributing to the global goal of ending TB by 2035. This study therefore explored facilitators and barriers for completion of the diagnostic process among people with presumed TB in selected health facilities in Central Uganda.

## Methods and materials

### Study design and setting

We utilised an explanatory qualitative phenomenological study design to explore facilitators and barriers for completion of the TB diagnostic process. This was part of a study to evaluate the TB diagnostic process and linkage to care in Central Uganda. Quantitative findings of the evaluation showed that 28% of the people with presumed TB were lost before completing the diagnostic process [12]. We hence sought to understand the influencing factors for the loss of patients and facilitators for completion of the diagnostic process. Focus group discussions (FGDs), Key informant interviews (KIIs), and in-depth interviews (IDIs) were conducted. The Consolidated Criteria for Reporting Qualitative Research (COREQ) guided the study-*S1 File* [15].

The study was conducted in purposively selected four health care facilities in Central Uganda that is; St. Francis Hospital Nagalama (Private-not-for profit) and Kojja health center four (IV) in Mukono district (lower-level health facility); Kawolo Hospital in Buikwe District and Mityana Hospital in Mityana District. Selection of study facilities was based on level of facility (hospital or health center) and ownership type that is either public or private-not-for profit to ensure rich information. Health care services in Uganda are offered by both the government (public) and private providers. The health system is arranged in a tier-system including national referral hospitals, regional referral hospitals, district (General) hospitals and health centers, four (IV), three (III) and two (II) [16]. General hospitals serve districts and provide general medical and surgical services targeting a population of approximately 500,000 people. Health center IVs serve a county or municipality with a population of approximately 100,000 people. They provide emergency surgery and blood transfusion services in addition to all general medical services [16]. TB services are offered from health center III up to the national referral hospitals. More than half of the households, 72% in Mukono, 68% in Buikwe and 72% in Mityana districts are within 5 km radius to the nearest public health facilities [17]. Over half of the population, 56%, own mobile phones in the Buganda region of the country where the study sites, Mukono, Mityana and Buikwe districts are located as indicated in the recent National Population and Housing Census results of 2024.

In all the study facilities, both the health center IV and general hospitals, the TB diagnostic process runs from being screened for TB symptomatically to receiving test results. Systematic TB screening is done at the different points of service delivery using the Intensified Case Finding Form [13]. A patient is presumed with TB if they have any of the cardinal symptoms: persistent cough of two weeks or more, night sweats, fever of more than two weeks, weight loss and if a child, poor weight gain and previous contact with a person with TB. Patients with presumed TB are then sent for laboratory or radiological investigations depending on the presenting condition. Xpert MTB/RIF is the recommended first diagnostic test in Uganda but also microscopy, urine lipoarabinomannan (LAM) and chest X-ray are used [13]. Patients' results are then sent to the point of service delivery that referred the patient. Patients are expected to return to the service delivery points and collect the results. Patients either receive results that same day of testing or are asked to return another day and check for their results hence they make several trips to the health facility until the results are ready. Diagnosed patients who test positive for TB and did not receive same day results are often called back to the health facilities through phone calls and initiated on treatment through the TB clinics [12,14]. Calling of patients is however inconsistently done due to resource limitations. Patients who test negative for TB are rarely called back; they are expected to return on their own whenever ready to collect the results. Patient demographics, address, telephone contacts and clinical details are recorded in the registers at every service delivery point for tracking and accountability purposes [13].

### Study population, sample, and data collection

We conducted 25 IDIs among patients identified with presumed TB in the study facilities. Participants were selected from the presumptive TB registers at each health facility by the research team supported by health workers. We included patients who were aged 18 years and above and were presumed for TB in the past one month. Purposive sampling was

used to select participants for the IDIs based on their completion status of the TB diagnostic process (completed diagnostic process for TB within 30 days from when they were presumed for TB/not completed at all). Selection ensured representation of both males and females. Conducting IDIs among patients who had completed the diagnostic process and those who had not and with the different sexes was to ensure rich and diverse perspectives of the findings. People with presumed TB who reported being critically ill were excluded. The IDI interview guide was developed containing open-ended questions on facilitators and barriers to completion of the TB diagnostic process as well as suggestions on increasing completion *(S2 File)*. Some of the questions included understanding of the patients' knowledge on TB, experiences as they navigated the TB diagnostic process, facilitators and barriers to submitting a sample for testing and collecting of results. The interview guide included probes to elicit more responses from participants but also to guide the interviewers on making checks for responses of previous participants to ensure credibility of the findings. The IDI guide was translated to Luganda, the most common language in the study area and pretested in one health facility in Wakiso district to ensure it yields the data needed. To limit bias in the study findings, the pretest was conducted in one health facility in Wakiso district, which had similarities with the study sites in Mukono, Mityana and Buikwe districts such as being in Central Uganda and having both urban and semi-urban catchment populations. The tool was then revised based on the feedback obtained from the pre-test to improve on the clarity of questions and include more probes. Interviews were conducted in *Luganda* and lasted about 40–60 minutes.

Six FGDs were also conducted with TB patients aged 18 years and above who had been on treatment for one month to understand their experiences as they went through the diagnostic process after being identified with presumed TB. The selection of one month was to limit the recall bias from the participants. The FGDs were organized to be sex-specific groups. This is because TB is a highly stigmatized disease hence participants in the opposite sexes would not be comfortable inadvertently disclosing their TB status and sharing experiences leading to potential biased responses. Participants were purposively selected by the research team from the TB unit treatment registers at each health facility based on the inclusion criteria with the support of health workers. A focus group discussion guide was developed containing open-ended questions and probes on facilitators and barriers to completion of the TB diagnostic process as well as suggestions on increasing completion *(S2 File)*. The discussion was conducted in *Luganda* by a trained researcher and lasted between 50 minutes and 1 hour.

Names of eligible participants for both the IDIs and FGDs sampled from the presumptive and TB unit treatment registers were handed over to the TB facility linkage officers who contacted the patients through phone calls using the contacts recorded in the registers at the health facilities. The TB facility linkage officers were introduced to the study and taken through the objectives and procedures for inviting participants for the interviews. During the phone conversation, they introduced the study to the potential participant as guided by the research team and invited the participant to come to the health facility for the interview if willing to participate. Selected patients whose phone contacts were unreachable at the time of data collection were contacted through the next of kin whose contacts are also recorded in the registers.

In addition, 20 KIIs with TB focal persons, heads of outpatients' departments (OPD), HIV/ART clinics, in-patient departments, laboratories and TB community linkage officers were conducted. The KIIs were selected purposively based on those who often interact and offer services to patients with presumed TB and those diagnosed in each study facility. That is, health workers who either provide TB screening or health education or counselling or testing or treatment to those diagnosed. Health workers who were on leave during the time of data collection were excluded from the study. We developed the KIIs interview guide containing open-ended questions with probes on experiences providing services to patients with presumed TB right from the point of screening to initiating treatment if diagnosed *(S2 File)*. We also explored their perspectives on likely facilitators and barriers to completing the TB diagnostic process and suggestions on increasing completion. Interviews with the health workers were conducted mainly in English.

The three methods of data collection that is; FGDs, KIIs and IDIs were used to enable triangulation of findings from the different methods and participants so as to obtain more rich data. All interviews were conducted between 18th April and

7th June 2023 by two trained qualitative researchers (RN and VK), male and female who hold a master's degree in Public Health and Sociology respectively. The two researchers were involved in the design of the study and drafting of the interview guides *(S2 File)*. A two-day training on data collection was conducted for the interviewers together with the rest of the study team to ensure quality of the data to be collected. The training covered basic information on TB and the diagnostic process, ethics of conducting research, skills and procedures in conducting qualitative interviews and an overview of the interview guides. All interviews were conducted at the study health facilities in a place deemed safe, private and comfortable for both the research assistant and respondent and there were no non-participants during the interviews to ensure privacy. All interviews and discussions were audio recorded after obtaining written informed consent from the participant and notes taken by a trained note taker. Meetings were held at the end of each day between the qualitative researchers who were conducting interviews and other study investigators to check for adherence to the data collection standard operating procedures and to discuss emerging codes and themes about completion of the TB diagnostic process.

## Data management and analysis

Audio recordings were transcribed verbatim in English by the researchers who conducted the interviews (VK and RN). The transcripts were cleaned and checked against the original audios to ensure quality and that complete data is available for analysis.

Thematic analysis was done following the steps suggested by Braun and Clarke [18]. Two researchers (VK and RN) experienced in qualitative research read through selected transcripts to familiarise themselves with the data. The researchers then read through the transcript the second time to develop codes independently using an abductive approach. For confirmability, the independent lists of codes from the two researchers were reviewed by other core study team members (EB, RN and LA) to assess agreement. Any discrepancies were clarified and resolved by comparing each coder's results with raw data until consensus was reached. The codes were then applied for line-by-line coding using ATLAS.ti software (version 6). The codes were analysed to formulate themes *(S3 File)* which were further refined guided by the constructs of the socio-ecological model and researcher's experiences during data collection. The socio-ecological model is based on the notion that behaviours are as a result of the complex interaction between humans and the environment where the two influence each other. The model conceptualizes this influence at different levels of; individual, interpersonal relationships, community and policy by Bronfenbrenner, 1994 [19]. Findings from this study were shared back with participants during an insight workshop for their feedback which further guided the interpretation of the results. The research team triangulated data from all interviews at analysis to improve on the credibility of the findings. Relevant quotes were selected to represent the main themes. We conducted methods and data source triangulation where we used various data collection methods that is; IDIs, KIIs and FGDs. These methods further included varied participants to ensure collection of data from different sources. These included participants who had completed the TB diagnostic process, and those who had not completed at all or initiated TB treatment if diagnosed for the IDIs; TB patients who had recently initiated treatment for the FGDs and TB health care workers. Data from these different methods and participant categories were coded independently by the researchers and later harmonized in the analysis meetings with all study investigators.

## Ethical considerations

Makerere University School of Public Health Research and Ethics Committee granted approval for the study *(Protocol number:799)*. The study was further registered with the Uganda National Council of Science and technology *(HS993ES)*. We obtained written informed consent and all participants signed the forms before participation in the study. All participants consented to be audio-recorded during the interviews and discussions. We maintained confidentiality of all data obtained by limiting access of the data to only the study investigator and data analysts. People with presumed TB who did not complete the diagnostic process, those who still had symptoms suggestive of TB and those diagnosed who had not started treatment were referred to the TB focal persons at the study sites for further management. Field researchers

practiced infection prevention and control measures at all times during the data collection to protect themselves and the participant against COVID-19, which measures also doubled for protection against acquiring the TB infection. Some of these measures included masking, handy hygiene with an alcohol-based sanitizer and keeping a social distance of 2 meters (*S4 File*).

## Results

### Characteristics of study participants

Out of 25 in-depth interview participants interviewed, 15 were female and almost a third aged between 45 years and above. Almost half (11/25) were farmers by occupation. We conducted six focus group discussions with 42 participants ranging between 6–8 per group, an average of seven participants. Most participants of the focus group (14), were aged 31–39 years of age and more than half (23), were females. Of the 20 key informants; three-quarters of them were female and most aged between 20–30 years (6) and 31–40 years (6).

Interviews from the patients (IDIs and FGDs) and health workers (KIIs) showed similar views on the facilitators and barriers to completing the TB diagnostic process, therefore the presentation of the results is integrated. Where necessary the similarities or differences across participant categories are highlighted. The findings are presented according to the categories of individual, health system and community factors. Six themes were obtained that is; i) individual level facilitators ii) individual level barriers, iii) health system facilitators iv) health system barriers v) interpersonal and community facilitators and vi) interpersonal and community barriers. Findings are summarized in Fig 1.

### Facilitators for completion of the TB diagnostic process

Three themes with corresponding sub-themes emerged under facilitators for completion of the TB diagnostic process: i) individual facilitators (persistent symptoms and the desire to get better, obtaining same day results and prior TB knowledge) ii) health system facilitators (caring health workers, support from cough monitors and facility linkage officers,

|  | Individual level | Health system level | Interpersonal and community level |
|---|---|---|---|
| **Facilitators** | -Persistent symptoms and the desire to get better<br>-Obtaining same day results<br>-Prior TB knowledge | -Caring health workers- positive interaction with patients<br>-Support from cough monitors and facility linkage officers<br>-Designating laboratory personnel for TB samples<br>-Calling of patients to collect results | -Social support |
| **Barriers** | -Inability to produce sputum<br>-Poor quality sample<br>-TB and HIV Stigma<br>-Fear of TB treatment<br>-lack of transport to return to the health facility | -High sample load<br>-Long turnaround time<br>-Long waiting time<br>-stock out of testing supplies<br>-Unclear appointment for collection of results<br>-Poor health worker attitude<br>-Inadequate patient contact details | - TB and HIV stigma |

**Fig 1. Results framework based on the socio-ecological model [19].**

designated TB laboratory personnel and calling of diagnosed patients to collect results) and interpersonal and community facilitators (social support).

**Individual level facilitators**

Almost all IDI participants who completed the TB diagnostic process and those in all FGDs, mentioned that experiencing persistent TB symptoms and the desire to get better was a facilitator for completing the diagnostic process. Participants stated that the pain alone from the persistent symptoms was a motivation to go through the whole process of getting a diagnosis and initiating treatment if diagnosed with TB in order to get better. In some instances, participants mentioned taking medication either from private clinics or pharmacies to get relieved of the symptoms which still persisted thus prompting them to get tested for TB. Persistent cough with chest pain was the commonly mentioned symptom mentioned by most participants, cutting across females and males. Relatedly, compared to females, more males expressed coming to the health facilities when already in too much pain hence motivated to complete the diagnostic process.

*"It was the pain that I was going through that motivated me to do everything that the health workers asked me to do. I always had fevers, would cough a lot and had been on treatment for a long time with no change…."* **39-year-old female IDI participant, completed the diagnostic process**

Obtaining same day results: few participants mentioned receiving results on the same day of being presumed for TB and attributed their completion of the diagnostic process to that. They said it was because they were able to wait and get their results without the need to return to the health facility which was associated with more transport costs, need for time off work and potential stigma from repeat visits to the health facility. Obtaining same day results was commonly mentioned by participants in the private not-for profit health facility. One participant in a female FGD affirmed that if she had not received results that very day, she would never have come back to complete the TB diagnostic process for fear of being stigmatized by her family.

*"I feared people at home to know that I have TB…if I was not told that I have TB on that same day and also given drugs, I do not think I would have come back to the hospital"* **25-year-old female FGD participant, Hospital**

Having prior knowledge on TB was mentioned as another key facilitator to completing the TB diagnostic process. This was because the patients already knew about TB, the benefits of testing and wanted to know the results. This was commonly mentioned by IDI and KII participants. The knowledge was obtained through previous experience with people with TB either by nursing or living with them, having a history of TB and interaction with people with TB in the community. Compared to females, males were less knowledgeable on TB. Additionally, sensitization from health workers in previous health facility visits was mentioned as a source of TB knowledge for some participants. In one of the KII, a participant explains how prior knowledge motivates the patients:

*"There are clients that we get who come straight to be tested for TB. They do not wait to be screened. They see the signs and symptoms and come. We get those too a lot. So, someone like that easily complies with the processes… Sometimes the client has a TB patient at home, so when they start to cough, they turn up to be tested too."* **47-year-old female KII_TB unit, Hospital**

On the other hand, in all FGDs, it was mentioned that some patients don't submit the sputum samples and return to collect results because they have limited knowledge on TB. The knowledge gap was specifically on the benefits of testing, not knowing that TB can be cured and consequences of not getting treatment if one has TB.

## Health system facilitators

Caring health workers: The positive interaction with healthcare workers motivated patients to get tested for TB. This was mentioned by most IDI participants who completed the TB diagnostic process and in most FGDs. The care was reported in the form of counselling, talking to them politely, guidance on the next steps in the diagnostic process and giving them extra information on TB. The care from health workers was also described to motivate patients who were diagnosed with TB to proceed further and start treatment. This was pointed out mainly by males and adults above 30 years across all the health facilities.

> *"According to what you have gone through while suffering from what you don't know, you can come here and ask for the tests yourself. The health workers are caring in a way that when you come, they immediately attend to you. The health worker's attitude and how they talk to us motivates us. These health workers always help to show us what to do and where to go."* **51-year-old male FGD participant, Hospital**

From the health system perspective, the care and support from cough monitors (health workers designated to assist in cough monitoring for a sputum sample from people with presumed TB) and TB facility linkage officers enables sensitization, symptomatic screening of patients and patient follow-up for samples and treatment initiation as mentioned by most KIIs. In the KIIs, it was noted that patient screening and follow-up would have been challenging with the heavy workload and low staffing especially in the hospitals. Having these support staff was therefore viewed as a strong pillar for supporting patients to complete the diagnostic process since they are mainly designated to do that with limited competing work. Furthermore, staff from the TB departments were described by all laboratory personnel as caring, supportive and responsible. They described their support in; sample collection, following up submitted samples for results, calling back patients (those with poor quality samples or whose samples were lost) and sometimes collecting the printed results and dispatching them to the requesting departments. This was mentioned to reduce the work load of the laboratory staff thus reducing the turnaround time of the results. In one health facility, staff from the TB departments were mentioned to support in sample collection from patients at the different service delivery points to reduce on the waiting time at the laboratory. One key informant mentioned how the support from the TB focal person helps them not to miss out on testing any patient;

> *"There is a close link with the TB personnel who really support us a lot...The good thing is that the TB focal person for the facility is really active. So, he always comes in and then checks on how many patients were worked on, what are the details, what could be the results."* **36-year-old female KII_laboratory, Health center IV**

Having designated laboratory personnel to run TB samples was mentioned in most KIIs as a strong facilitator for having a diagnosis for patients. These designated personnel prioritize the TB samples in addition to the other laboratory workload and ensure that all samples received are run on time and sends back results to the requesting departments. It was noted that these personnel also take the initiative to alert the requesting unit and TB focal person once a patient's result comes out positive. This is to ensure that the patient is called back to the health facility to initiate treatment. In two health facilities, it was mentioned that this laboratory person prioritizes patients with presumed TB for sample collection to reduce their waiting as a way to motivate them to complete the diagnostic process. Additionally, one key informant, a laboratory manager, mentioned that not all staff want to process and analyse TB sputum samples because they find them unpleasant. She explained that previously, most of the laboratory staff would leave the sputum samples unanalyzed for her to process when on duty. This prompted her to designate specific personnel to analyze these samples so that patient's results are not delayed.

Calling of patients diagnosed with TB to collect results also facilitated almost all participants who were diagnosed to collect results and start treatment. In the majority of the KIIs, participants said they always call back patients who are

diagnosed with TB to return to the health facilities and initiate treatment immediately. This happens in cases where the patient had left the health facility without collecting their results. Health workers contact patients by telephone using the contact details recorded in the health facility registers or those of the next of kin if the patient is unreachable on their personal telephone. Participants in some of the IDIs and all FGDs also said the same, that the phone call helped them to come back to the facility, collect results and initiate treatment.

*"The health worker called me directly and told me that the results from my sputum show that I have TB and also asked me if I can go there on Monday and start taking drugs. I was going through a lot of pain. I had coughed for weeks… I was looking forward to Monday to go to hospital and start my medications."* **39-year-old female FGD participant, Hospital**

However, some KIIs stated that calling of patients was sometimes challenged by no airtime to make the phone calls and poor record taking of the patient contact details. It was noted that some health workers do not record the contact numbers of the patients or record incomplete numbers. On the other hand, other KIIs pointed out that patients submit wrong or incorrect contact numbers when they feel they don't want to be contacted. This was mentioned to be common among young people and those who stay in distant places.

*"The other problem I have seen is amongst the youth and the people on islands, some of them forge numbers (telephone contacts). They give us forged telephone numbers. There is a client that we received I think in 2021 from Buvuma (an island). The lady turned TB positive, we tried to call and the phone was not available. It was not available on the network. So that is a problem too."* **23-year-old male KII_TB Unit, Hospital**

### Interpersonal and community facilitators

Receiving social support was mentioned by most IDI participants and few FGDs as a key factor that encouraged them to get tested for TB, collect results and initiate treatment when diagnosed. The support came mostly from parents, spouses and close relatives in the form of accompanying them to the health facility, advising and counselling them to get tested, and giving them money for transport and food while visiting the health facility. The social support was more pronounced among females and the young people under 35 years of age as narrated below;

*"My mother encouraged me to come for testing to see what is wrong with them. She also counselled me and told me whatever the results I will not be the first."* **23-year-old female IDI participant, completed diagnosis**

### Barriers for completion of the TB diagnostic process

We identified three themes under barriers for completion of the TB diagnostic process: i) individual-level barriers (inability to produce sputum, TB and HIV stigma, lack of transport to return to the health facility, fear of TB treatment) ii) health system barriers (long turnaround time for results, high sample load, stock out of testing supplies, unclear appointment for collection of results, poor health worker attitude, inadequate patient contact details) and interpersonal and community barriers (TB and HIV stigma).

### Individual-level barriers

A few IDI participants who did not complete the TB diagnostic process mentioned that they were unable to provide a sputum sample into the laboratory specimen container for testing. They mentioned trying all they could to produce the

sputum but were unable and eventually went home. Very participants mentioned that they returned to the health workers for advice when they were unable to produce the sputum. These were advised to get an early morning sample while other participants said they simply left the health facility without consulting. For some participants, they were still unable to produce the early morning sputum sample and others were successful. Those who were unable to produce the sputum completely mentioned not returning to the health facility for fear of being reprimanded by the health workers if they came back without a sample. Because of the fear to go back to the health workers, patients ended up either submitting a poor-quality sample or disappearing from the health facility.

*"I wanted to know the results truthfully but I did not have sputum. I do not know what I put in there. I coughed, coughed, and coughed…but I did not have it. When they asked for it (sputum), I did what I did and spat in the container but it was not sputum."* **39-year-old female IDI participant, lost to follow-up (LTFU)**

A few KIIs explained that some patients are unable to produce sputum because health workers rarely tell patients the techniques of sputum production during sensitization and counselling sessions. It was noted that patients are also not informed of other available tests in instances where one is unable to produce sputum which causes patients to disappear from the health facilities. It was further mentioned that limited sensitization also leads to poor quality samples from patients with some submitting either saliva or an insufficient sample. A key informant narrates below how patients do not return to the health workers once they are unable to produce sputum.

*"There are some patients when they are told to collect sputum, they find it difficult to get that sputum, so when they fail, they just give up. When they are asked to collect sputum, they go in that area where they are supposed to collect the sputum from and when they fail, they just lose hope and sometimes they are not explained to the other options. So, they just give up and go away."* **34-year-old male KII_TB Unit, Hospital**

TB and HIV related stigma was reported to lead to loss from the TB diagnostic process. The drivers of the stigma were; i) fear of association with HIV/AIDS, ii) fear of social exclusion and iii) fear of spreading TB to the rest of the family members. Participants in the IDIs and FGDs feared being labelled as people living with HIV because of the community understanding that whoever has TB also have HIV. They also expressed the fear of being stigmatized and socially excluded in the community by family members if they found out that they were being presumed for TB. Hence, they feared to be isolated, gossiped about, being seen using TB services and taking medication. The stigma was reported across all ages and sexes, and by participants in all IDIs and in all FGDs.

*"It is because of fear. Personally, I was initially afraid as well. I imagined what it would be like if I were seen taking TB drugs, and my mom telling me not to take them from here. I also thought about how people would gossip about me if they saw me taking those drugs. So, I decided to leave it up to them and not take the drugs because I was worried about how people would perceive me."* **27-year-old female FGD participant, Hospital**

The barrier of TB and HIV stigma was also attested to by key informants who mentioned that the community associates TB to HIV/AIDS which discourages people with presumed TB from testing for fear of being labelled by the community as people living with HIV. A key informant in one of the hospitals explained how stigma makes patients not to test for TB:

*"Stigma makes them disappear because most people think that when you have TB you are HIV positive…So these patients fear seeing them in the line of people testing for TB and HIV…that kind of fear causes the patients to get lost and fail to complete the diagnostic process. The person is asked to produce the sample and they instead choose to leave the facility"* **43-year-old male KII TB Unit, Hospital**

The TB/HIV testing policy being implemented in the study facilities was mentioned to further increase the stigma which deters away some patients when presumed for TB. Majority of the participants in KIIs and IDIs and in few FGDs said patients know that they will not only be tested for TB but also for HIV and this scares them so much. The main fear was getting HIV positive results once tested followed by daily swallowing of drugs for the rest of their lives which scared them away. Participants stated that some community members thus prefer to live in ignorance of their HIV status than face this reality of swallowing drugs daily. The fear of being tested for HIV was commonly mentioned among males and participants under 40 years of age.

> *"Some people are told to go and test but they refuse because they fear and they know they are not only going to be tested for TB but they also have to be tested for HIV and this thing scares them so much"* **36-year-old male FGD participant, Hospital**

Lack of transport to return to the health facility either to submit a sample or collect results from the test was also reported as a barrier for completion of the TB diagnostic process. This was mentioned by participants who did not get a spot sputum sample and needed to take back the sample the next day and by those who did not collect results or initiate treatment on the day of the health facility visit. Participants reported that they do not receive results on the same day they are presumed for TB hence the need to return to collect results or initiate treatment which requires money for transport. This was commonly mentioned by females and those who reside in areas far away from the health facilities. Participants who stay far said it is worse for them compared to those who stay in nearby places. Close to half of the participants across IDIs, KIIs and in all FGDs mentioned the lack of transport as a major barrier.

> *"The distance and the transport fees made it difficult for me. If it were nearby, I would have returned even that very day…. But the distance is long and you have to first gather some money before you can be able to come back to the facility.".*" **50-year-old male IDI participant LTFU**

The fear of TB results and treatment was mentioned in all FGDs and by most IDI and KII participants as a key contributor to loss before completing the TB diagnostic process. It was noted that generally some people fear taking drugs hence they will disappear from the health facilities before knowing the results. The fear of taking TB drugs is further amplified by the TB and HIV stigma with participants fearing to be seen taking TB drugs and be mistaken for swallowing HIV drugs. Other participants mentioned knowing that TB treatment is for six months hence see it as a long time, which makes them not to proceed to test for TB preferring to live in ignorance of their status. Other participants who had seen the drugs perceived them as big tablets hence feared swallowing them. All these perceptions on the TB drugs and duration were reported to discourage patients from testing for TB.

> *"Some people fear to know the results and also fear taking the drugs. Some people have fear within themselves and also do not imagine themselves taking the TB drugs especially when they have seen the drugs. Tablets are too big and are taken for a long time. Some people feel sorry for themselves and say that they do not want to take drugs. This makes them not to come back."* **52-year-old male FGD participant, Hospital**

More so, people living with HIV were cited to fear the potential pill burden in case the results come back positive for TB. This is because they are already swallowing ARVs daily and perceive the addition of other daily TB drugs as cumbersome hence they shy away from the diagnosis. This causes them to not test for TB or even collect their results.

> *"Another thing that makes people to not want to complete the process (TB Diagnosis) is the fact they are already taking HIV drugs. They imagine how they will be able to take both drugs concurrently and so they will decide to stay on the*

*HIV drug only…The fact that the TB drugs are also taken for a long period of six months and yet the HIV drugs are already for a lifetime makes them to stay away from completing TB diagnosis. They just have hope that they will some-how recover from the TB without drugs yet that is not possible".* **40-year-old male FGD participant, Health center IV**

**Health system barriers**

High sample load from lower-level health facilities was mentioned in the two public hospitals as a barrier for same day analysis of samples. These hospitals receive many samples from different referring facilities in addition to those collected from the patients seen in their facility departments. This was further complicated by the type of Xpert MTB/RIF machines available in the facilities which can only analyse few samples at a time. This led to a long turnaround time for results and consequently long waiting time for the patients. In most cases, they asked the patients to return and collect results either the next day or after some days which led to some of them getting lost from the diagnostic process.

*"We work on samples from lower facilities and in a day, we can work on thirty (30) or more than that throughout the district. This gene expert takes one hour to run, just running and when it comes to sample collection, we can give it like three (3) hours in total. You find that time for patients to wait is long. It is too hard for them to wait...As I have told you, when they go back to the village, coming back is a problem."* **30-year-old male KII_laboratory, Hospital**

Relatedly, long waiting time for results was mentioned by over half of the participants in IDIs, KIIs and all FGDs. Patients reported a long waiting time either on the day of sample submission when told to wait for results that very day or long wait-ing in days when one is told to return because the results will not ready on the same day of sputum submission. The long waiting time was said to lead to frustration and one making a decision to either return or not return to collect the results. For patients who are told to return on another day and collect results, it was mentioned that some health workers attempt to give them a date to return with the hope that results will be ready then. However, it was not always guaranteed that the patients will find results on the given date. Some participants mentioned the possibility of not finding results on that day they are told to return and collect them. This therefore meant more waiting time and the need for another return visit to the health facility. Participants narrated that every return to the health facility necessitated time off work and more transport money beyond that which was originally planned at the first visit. This was therefore seen to make completion of the TB diagnostic process tedious and costly.

*"The long waiting time for results also makes them disappear. For example, those who have given in their sample on Tuesday are told to come for their results on Friday. But remember he may come on Friday and be told to come back on Monday. All that is money wasted on transport."* **44-year-old female KII_TB Unit, Hospital**

On the other hand, some participants mentioned not being given clear appointment dates when to collect the results which left them in uncertainty causing them to make random visits to the health facilities before obtaining results. Unclear appointments for collection of results coupled with poor communication was reported by majority of the IDIs and in three FGDs as a barrier. This meant patients making several back and forth trips to the health facility checking for readiness of results which was mentioned by some participants to expose them to stigma and doubt by the people around them.

*"They did not call me and they also did not give me a date so that when the date reaches, I go back. But they did not tell me so I didn't know. When they did not give me a return date or call me, I went silent…what scared me was fre-quenting the hospital and what people would say and think about it, especially my mother. Had they called me, I would have come back."* **37-year-old female IDI participant_LTFU**

*"I thought that they would give me the results on that day, but they told me that they would call me and tell me when to return…They did not call me and they also did not give me a date. When they did not give me a return date or call me, I went silent."* **52-year-old female IDI participant_LTFU**

Stock out of testing supplies at the health facilities also hindered patients from completing the diagnostic process. Laboratory staff in the hospitals mentioned a common stock out of Xpert MTB/RIF cartridges which forces them to halt sample collection and processing. More so, a few participants said that some tests not being available at the health facility increases the likelihood of patients being lost from the process. In such cases, patients are sent for testing outside of the health facility, which was often associated with costs. In that process, the patients do not return to the health facilities either because of the cost or they are unable to get time and transport to return to the health facility. Chest x-ray was the commonly missing test in the public facilities and patients were often referred to the private facilities with the machine which came at a cost.

*"It (Chest x-ray) is not so common because it has got a cost and it's not within our health facility and area, so one has to travel to Mukono town for x-ray. So then coming back with results is not common. It may be a handful of them who come back."* **29-year-old male KII_TB unit, Health centre IV**

Poor health worker attitude towards TB was also reported as a barrier when screening and handling patients with presumed TB in all FGDs and in some IDIs. Some KIIs especially in hospitals reported that health workers handle these patients with a negative attitude in terms of the way they screen them, send them for sputum samples and respond to them when they ask questions or have made an error in any of the health facility processes. Relatedly, some patients mentioned being stigmatized by health workers who shout at them to wear masks when coughing and to extend far when talking to them. This attitude made the patients feel unwanted hence feared to return to the health worker when asked to submit the sputum sample.

*"TB is so painful. The health worker was asking me questions and I could not answer the questions. She asked me if I did not have a mouth to speak. I had the mouth, but was unable to speak. I did not know what to tell the health worker."* **41-year-old female IDI participant, completed diagnosis**

Health workers also acknowledged the fact that the negative attitude of their fellow staff demotivates patients to complete the diagnostic process. Patients were mentioned to seek care when in pain and after trying many different options to get healed but in vain, including going to the traditional healers. This was mentioned as costly and frustrating for patients which requires handling them with care when they finally come to the health facilities.

*"The health workers bad and negative attitude towards the patient scares them away. At times these patients come here when they no longer love themselves because of the pain. Actually, some sacrifice their stuff to witch doctors to get healed so they need to be handled well."* **55-year-old male KII_TB Unit, Hospital**

Inadequate patient contact details: some of the key informants noted poor documentation of patient tracer information such as telephone contact and home address as a key barrier for receiving results and initiating treatment. Participants described this as either contacts not being collected at all or contacts available but incorrect. This tracer information was mentioned to aid health workers with patient follow-up especially when a patient's results are positive for TB and contact tracing as below;

*"One of the reasons (for the loss of patients) is poor documentation. So, you presume someone and you don't write their phone numbers and not even their next of kin phone number. Those people you can hardly find them… You find*

*that all you know about that person is where they live and the area is very big. Even people have the same names, so some people are lost in that way.”* **38-year-old male KII_ OPD, Health centre IV**

## Discussion

This was a qualitative exploration of facilitators and barriers to completion of the TB diagnostic process among patients with presumed TB, those diagnosed with TB and health workers providing TB services in selected health facilities in Central Uganda. Key facilitators for completion of the TB diagnostic process included; social support, obtaining same day results, prior TB knowledge, caring health workers (positive interaction with patients who were helpful, caring and compassionate) and calling of patients to collect results. Barriers were; long turnaround time, lack of transport to return to the health facility, TB and HIV stigma, unclear appointment for collection of results, inadequate patient contact details and negative health worker attitude.

Social support from family and community members were mentioned as a key facilitator to completing the TB diagnostic process. This support was commonly mentioned from the significant people in patient's lives who have a form of influence such as spouses, parents and siblings. This was found in studies elsewhere where family members escort people with presumed TB to testing facilities, others provide transport means and offer words of counsel and encouragement to them to visit the health facilities [14,20,21]. The social support is crucial given that TB is a highly stigmatized disease which could deter one from testing if not supported and motivated by a significant person in their life [22,23].

Obtaining same day results facilitated completion of the TB diagnostic process. This is because the patients got the diagnosis there and then and did not require more time off work and transport to return to the health facility, which were often cited as hard to get. This was further attested to by patients who did not complete the diagnostic process mentioning failure to obtain same day results as a barrier. Although the goal of all TB programs is to have patients with presumed TB obtain their results on the same day of testing, this is not realized yet [24]. Evidence from a systematic review and meta-analysis in high-TB burden countries shows a health system delay for TB diagnosis of 14 days [25]. More so, our formative study showed that over half of people with presumed TB who conducted the TB investigations required, did not receive their test results on the same day of being tested [12]. Failure to provide same day results was linked to high sample load, stock out of supplies in this study and studies elsewhere showed additional factors of inadequate laboratory personnel, and low laboratory capacity [14,26,27]. Delay to provide a TB diagnosis exposes the community to possible infections and increases morbidity and mortality in people with TB [8,28–30].

Relatedly, long turnaround time and waiting time for results in this study led to the need for return visits to the health facilities to collect results which contributed to the loss of patients before completing the diagnostic process. Return visits required transport money, taking time off work and exposed participants to potential TB/HIV stigma due to the frequent hospital visits. This is similar to what other studies have found in Uganda and elsewhere [6,14,21,22,31,32] where long turnaround time for results and need for repeat visits leads to loss of patients from TB care. The need for repeat visits to the health facilities increases health care costs for the patients which may derail progress towards achieving universal health coverage and the goal of eliminating catastrophic costs for TB services by 2035 [33]. Uganda is already struggling with high pre-diagnosis catastrophic costs of 30.6% [34] hence this could further push patients to poverty. There is therefore need for increased commitment for both financial and human resources to enable issuance of TB results to patients on the same day of the health facility visit.

Unclear appointment on the day for collection of results limited the ability of patients to complete the TB diagnostic process. This was because the patients remained unsure of when to return to collect results, were uncertain if they would be contacted when results are ready but also those who were promised to be called mentioned not being contacted when results were ready. Studies elsewhere showed the same with patients citing poor communication as a barrier to completion of the diagnostic process [35]. Given the structural and personnel inefficiencies in low-resource settings to provide

same day TB results, emphasis should be placed on improving communication with patients once results are ready. From this study, those who were called back to collect results via phone calls, especially those who were diagnosed with TB, mentioned this as a key facilitator. More so, there is overwhelming evidence on the effectiveness of phone call and SMS tools in improving health care seeking [36–38]. However, health workers indicated inadequate patient contact details as a barrier even when there was an opportunity to contact the patient to return to the health facility. Lack of patient contact details has been seen in previous studies as a key contributor to loss to follow-up of patients with presumed TB [12,39–41]. Improving patient-provider communication therefore requires comprehensive strategies comprising of proper documentation and availing the communication tools and resources.

TB and HIV related stigma was a key barrier to completing the TB diagnostic process as has been found in other studies in Uganda, Kenya and across the globe [14,32,42,43]. This was mainly perceived stigma due to the misconception that whoever has TB has HIV hence the fear was more on being associated with HIV. This stigma was further complicated by the WHO and Uganda national guidelines on TB/HIV testing where patients are expected to test for TB and HIV at the same time which breeds stigma and fear especially if the HIV testing is offered in a facility that serves people living with HIV [44–46]. HIV remains highly stigmatized despite on-going efforts which is now consequently affecting the opportunistic infections complicating their control and management [23,47]. More so, there was fear of being socially isolated once one is known to be presumed or diagnosed with TB. Literature shows that community members often do not want to interact with people with TB for fear of being infected, even when they do, they express and show discomfort [23,48]. This social isolation reduces the patient's self-esteem and self-efficacy to seek care leading to delayed diagnosis, treatment initiation, poor adherence to treatment and poor mental health including suicidal thoughts [49–52]. A study in LMICs revealed that poor mental health combined with TB infection leads to worse outcomes than TB infection alone hence the need for urgent action [53]. Sensitization efforts should focus on improving community knowledge on TB and dispelling TB misconceptions to reduce the stigma. This will consequently encourage social support to the patients as has been seen in previous studies where patients who received social support were less likely to experience stigma [49,54,55].

Prior knowledge on TB obtained through health education or interfacing with people with TB was mentioned to increase one's ability to complete the TB diagnostic process. This was because they already knew more details about TB such as the benefits of early testing and starting treatment and the fact that TB can be cured and some feared they may have been exposed hence motivated to test and know their status. These findings are similar to studies elsewhere that show previous history and experience with a person with TB increases the likelihood of testing for TB [20]. Previous studies elsewhere have found that limited awareness of TB is associated with delays in healthcare seeking [27,35,43]. This therefore calls for strengthening of community and facility health education programs on TB.

Although the attitude and behaviour of health workers has been criticized over the years as deterring patients to seek care in health facilities, our study found otherwise [56–59]. Health workers who offer services to people with presumed TB were described as caring indicating a positive interaction finding them helpful, caring and compassionate. This motivated most participants to return to the health facility and complete diagnosis. This finding is similar to recent studies that have showed high patient satisfaction with health worker behaviors and attitudes [60,61]. This may be due to increased awareness and engagements with health workers towards a patient-centred care approach that emphasizes improved communication with the patients which makes them feel cared for [62,63]. Furthermore, it maybe because TB is a highly stigmatized disease with a long-term treatment period hence the health care workers show empathy to the patients. Previous studies have showed a positive correlation between improved knowledge and positive attitude hence the need for continuous engagement of health workers for attitude change to be able to provide better services to patients [64–67]. On the other hand, a few participants were dissatisfied with the health care workers attitudes especially during screening of people with presumed TB hence the need for more behavior change programs among health workers.

A key strength to our study is the triangulation of findings from different interview methods that is; IDIs, FGDs and KIIs. More so, we triangulated data from different levels of health facilities to explore these factors from a health system

perspective. This triangulation provided rich information on understanding completion of TB diagnosis from perspectives of the patients and health care workers in the different settings. This information may inform planning for strengthening provision of TB services in the country. We conducted sex-specific FGDs for homogeneity but also to mitigate the likely gender-related TB and HIV stigma among participants which promoted a conducive environment for free and open participation.

However, there might have been recall bias from participants in the FGDs who had initiated treatment within one month. They might have inaccurately recalled the events surrounding their experience during the TB diagnostic process. This was however mitigated by purposively selecting most from those who initiated within two weeks to reduce the recall bias. Further, despite efforts to maintain objectivity, our social identities, prior experiences and perceptions may have shaped interactions with participants during interviews and the interpretation of data during analysis. This was minimized by using the same interview guides throughout and disseminating preliminary findings among different stakeholders for their insights but also triangulating data from the patients and health workers. In addition, adoption of the socio-ecological model comes with challenges in fitting the results accurately to express the dynamic interplay between factors within and at different levels, with some levels not being well represented like the policy level.

## Conclusion

This study reveals persistent gaps in Uganda's TB response for people with presumed TB which need urgent attention to prevent more people from dropping out before completing the diagnostic process if the country is to achieve the 2035 End TB strategy. Obtaining same day results, social support, caring health workers and prior knowledge on TB facilitate completion of the TB diagnostic process while long turnaround time for results, TB and HIV stigma, poor health worker attitude and inadequate patient contact details contribute to the loss of patients. Addressing diagnostic and resources challenges to enable same day results, improving health worker to patient communication and enhancing social support from the community should be considered to enhance completion of the TB diagnostic process. The national policy for same day testing of TB and HIV should be accompanied with heightened community awareness and counselling to minimize the associated stigma that comes with HIV testing which was not anticipated by the patients.

## Supporting information

**S1 File. The Consolidated Criteria for Reporting Qualitative Studies (COREQ) Checklist.**
(DOCX)

**S2 File. Interview guides.**
(DOCX)

**S3 File. Analysis framework for factors for completion of the TB diagnostic process among patients with presumed TB in Central Uganda.**
(DOCX)

**S4 File. The study risk mitigation plan.**
(PDF)

## Acknowledgments

We extend our appreciation to Ms. Brenda Katusiime and the health workers in the study health facilities who supported the data collection and management processes. We also acknowledge the peer review from Dr. Arthur Bagonza, Dr. Rawlance Ndejjo and Mr Tonny Ssekamatte.

## Author contributions

**Conceptualization:** Rebecca Nuwematsiko, Lynn Atuyambe, Esther Buregyeya.

**Data curation:** Rebecca Nuwematsiko, Esther Buregyeya.

**Formal analysis:** Rebecca Nuwematsiko, Vicent Kasiita, Esther Buregyeya.

**Funding acquisition:** Esther Buregyeya.

**Investigation:** Rebecca Nuwematsiko, Vicent Kasiita, Esther Buregyeya.

**Methodology:** Rebecca Nuwematsiko, Lynn Atuyambe, Noah Kiwanuka, Esther Buregyeya.

**Project administration:** Rebecca Nuwematsiko, Esther Buregyeya.

**Resources:** Esther Buregyeya.

**Supervision:** Lynn Atuyambe, Noah Kiwanuka, Esther Buregyeya.

**Validation:** Rebecca Nuwematsiko, Lynn Atuyambe, Noah Kiwanuka, Angella Musiimenta, Elizeus Rutebemberwa, Esther Buregyeya.

**Visualization:** Rebecca Nuwematsiko, Lynn Atuyambe, Angella Musiimenta, Elizeus Rutebemberwa, Esther Buregyeya.

**Writing – original draft:** Rebecca Nuwematsiko.

**Writing – review & editing:** Rebecca Nuwematsiko, Lynn Atuyambe, Vicent Kasiita, Noah Kiwanuka, Angella Musiimenta, Elizeus Rutebemberwa, Esther Buregyeya.

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
