## [Decision Letter · Decision Letter 0]

24 Mar 2025

PGPH-D-25-00066

“If I was not told that I have TB on that same day and given drugs, I don’t think I would have come back to the hospital” Facilitators and barriers for completion of the diagnostic process among patients with presumed tuberculosis in Central Uganda

Dear Dr. Nuwematsiko,

Thank you for submitting your manuscript to PLOS Global Public Health. After careful consideration, we feel that it has merit but does not fully meet PLOS Global Public Health’s publication criteria as it currently stands. Therefore, we invite you to submit a revised version of the manuscript that addresses the points raised during the review process.

We look forward to receiving your revised manuscript.

Kind regards,

Suman Majumdar

Academic Editor

Journal Requirements:

Additional Editor Comments (if provided):

revision. Please focus on more clarity and specificity in the methods, as per reviewer 2.

Title: I support the use of a quote as the manuscript title, however, it is long. Please consider an alternative that can be shorter

Please review the language in the manuscript using the Stop TB language guide, as suggested by reviewer 2

Please address all reviewer comments, noting my additions below:

Introduction:

• Update quoted statistics to most recent (not 2022) – line 29

• Reference 15, should be referred to in the introduction; as well as the context of this study being the qualitative component of a larger program of work

Methods

• I assess the study methods as sounds, using standardised and appropriate approaches for thematic analysis and a socio-ecological model. Noting this when responding to reviewer 2’s comments, I do agree any potential bias needs to be acknowledged and articulated.

• Setting: what were the distances (and time) patients had to travel in the catchment areas to access the health facilities? Describe this as part of the setting

• A clearer explanation of whether there is a protocol or SOP for follow-up of new TB diagnoses – or it is up to clinicians to make phone calls / follow them up. What proportion of patients / the general community have a mobile phone in the health facility catchment areas (is it very common) ? (line 110-115)

• Why did the purposive sampling not include distance from health facility?

• No need to list the Braun and Clarke steps – just reference

Findings

• Please title this as “results” (line 214)

• Distance from health facility is a key characteristic to report

Reviewers' comments:

Reviewer's Responses to Questions

**Comments to the Author**

1. Does this manuscript meet PLOS Global Public Health’s publication criteria ? Is the manuscript technically sound, and do the data support the conclusions? The manuscript must describe methodologically and ethically rigorous research with conclusions that are appropriately drawn based on the data presented.

Reviewer #1: Yes

Reviewer #2: Yes

2. Has the statistical analysis been performed appropriately and rigorously?

Reviewer #1: Yes

Reviewer #2: N/A

3. Have the authors made all data underlying the findings in their manuscript fully available (please refer to the Data Availability Statement at the start of the manuscript PDF file)?

Reviewer #1: Yes

Reviewer #2: No

4. Is the manuscript presented in an intelligible fashion and written in standard English?

Reviewer #1: Yes

Reviewer #2: Yes

5. Review Comments to the Author

Reviewer #1: Reviewer comments

I appreciate the authors for the well written piece of work. I have a few comments which need to be addressed to enhance readability and improve the quality of the paper—most of them are minor comments.

Methods

1) Line 91-Write IV in full the first time its used and then proceed to abbreviate in the subsequent text.

2) Align your reporting of the study to recognized standards for reporting of qualitative studies such as the COREQ checklist. Use the checklist and attach it as a supplementary material.

3) Line 181-Indicated who participated in those meetings which were conducted at the end of the interviews.

4) Line 110-117: This section lacks references.

5) Line 135-137: Indicate if the site where interview guides were piloted is not the same as one of the study sites, as this might bias your findings if it’s one of your sites.

6) Line 139: Remove redundant statement……” common language in central Uganda”. You already stated under line 135/136 that Luganda is the most common language in the study area.

7) Ethics Line 206-212: Did this study put a mechanism to link patients presumed to have TB to diagnostic or treatment services? Indicate in the ethics section how they were supported to access these services.

Were there no concerns of infection prevention and control on the side of the researcher and cross-infection during FGDs considering that you were handling potentially infectious individuals?

Results

Line 260: Delete the redundant the word “participant” after FGD.

Reviewer #2: This article is a qualitative study including in-depth interviews, key informant interviews and focus group discussions exploring facilitators and barriers to completion of TB diagnostic processes in four health facilities in Uganda conducted in 2023. Findings highlight the need for addressing diagnostic gaps and challenges, including the provision of same-day testing results, improving social support mechanisms and healthcare worker to patient communication – which have been previously highlighted in the literature as important components of patient care in relation to diagnosis and linkage to care. I have reviewed the manuscript in detail and also considering the COREQ Checklist for the publication of qualitative research.

Major comments / questions

1. Regarding the use of a quote in the article title

1.1. Although it is common for a quote to be used in qual article titles, I have a question about why out of all facilitators and barriers this quote was chosen - and indeed out of all the factors why this aspect seems to be highlighted the most. I suggest considering whether the quote is essential, or whether potentially reductive.

1.2. If the authors choose to use the one quote in the title, I suggest explaining how / what methodology was used to select the one quote/aspect as representative of the data.

2. Methodology

2.1. Was there a methodological orientation underpinning the study? If so, it should be mentioned.

2.2. Given the long list of facilitators and barriers, did they ask participants which they thought were most important? This could also assist with structuring the findings to help assist program/policy makers.

2.3. Were any themes identified in advance, or all derived from the data? How was the triangulation conducted? Authors could consider explicitly describing their role in identifying themes, rather than suggesting that themes ‘emerged’ from the data.

2.4. Data in this study was collected through FGDs, KIIs and IDIs. It is not clear from the methods why these three approaches were chosen.

3. Reflections of researchers’ roles and potential bias

3.1. It would be beneficial to include any reflections from the researchers on their own role, potential bias and influence/s during formulation, data collection and particularly during the selection of data for presentation.

3.2. Under findings I recommend highlighting whether or not the topic was spoken about by both healthcare workers and patients for each theme/sub-theme, how the themes were selected; and if there were any minor themes not included.

4. Inclusion of interview guides in supplementary files

5. Sampling

5.1. More detail required on how purposive sampling was done - e.g. line 163 KIIs were selected purposively, based on what criteria? Type of HCW or only on sex and age?

5.2. Lines 123-125: ‘Purposive sampling was used to select participants for the IDIs based on their completion status of the TB diagnostic process (completed diagnostic process for TB/not completed at all)’ – does this mean all patients were eligible because all would either have completed or not completed at all?

5.3. Line 152 sampled - how were they sampled, by whom?

5.4. If purposively sampling for sex how did an imbalance occur (15 females/10 males)- were females more likely to agree?

5.5. How many people who were loss to diagnosis or loss to treatment were planned to be included? How many were included?

5.6. Is there any information on why / how many people did not accept to be interviewed for IDI and FGD? Or dropped out?

6. Findings

6.1 The results state in line 222 that the findings were similar for each of the three approaches, so the findings were combined. The next sentence then talks to the similarities and differences. How was analysis and combining or not of data originally planned - the methods state that triangulation of the analysis was planned line 201-202 – how is this different to what is stated in the results line 222?

6.2 The paper has a focus on same day results, but not all people with TB would be able to be diagnosed within the same day with current tests? Did the study only include people with microbiological confirmation? The suggestion of same day results as a solution seems to lack a more detailed nuanced explanation around the challenges of achieving this and the root causes. I think the findings overall need to be put into a clearer context. It is not clear whether there was anything found that was new or unexpected, and how the findings can be applied to help with Uganda’s TB response. It would be beneficial to narrow down and specify the contribution of the study to existing practice/policy and how this could be applied within the local context.

Minor comments and corrections

Language

- A few suggestions around language usage (i.e. ‘case’ in line 22, 'positive patient' in line 596, ‘failure to produce sputum’ coming across as ascribing blame to the individual) are given below, based on the Stop TB Partnership guidance https://conf2022.theunion.org/wp-content/uploads/2022/05/stbp_words-matter_screen-ready.pdf

Introduction

- Suggest the use of global data for 2023 as supposed to 2022

- There should be a definition of ‘TB diagnostic process’ and what completion of the process means

- Line 54 and throughout: ‘patient with presumed TB’ - consider the use of ‘person with presumed TB’

- Line 56: ‘Pre-diagnosis and pretreatment loss of patients facilitates continued spread of TB in the community, poor treatment outcomes for those infected and death’ – suggest re-wording ‘those infected’ for person/people with TB

- Lines 59-61: ‘This burden is shared across all TB-burden countries although it is highest in high-TB burden countries compared to low-burden countries’ - needs rewording for clarity

- Lines 71 and 73: these two sentences are relating the same information and can be consolidated.

- Lines 76-78: ‘Understanding factors that influence completion of the TB diagnostic process will provide insights into effectively implementing screening and diagnostic strategies thus contributing to the global goal of ending TB by 2035.’ – This claim needs to be revisited. The findings would help with gaining in-depth understanding of factors which are relevant in this specific context. Although findings may be transferrable to similar contexts, they should not be generalised.

Methods and materials:

- Line 86: ‘28% of the patients with presumed TB were lost before completing the diagnostic process and’ – this sentence is incomplete.

- A definition for ‘diagnostic process’ and what completion means should be added

- Lines 90-93 – please review use of round bracket/parentheses

- Line 103 – would be good to clarify which facilities are included in the sentence ‘In each of the health facility’ – does this include health centre III up to the national referral hospital?

- Line 106: ‘contact with a TB patient’ – consider re-wording to person with TB.

- Line 111: ‘Patients are expected to return to the service delivery points and pick the results.’ – unclear whether they pick up the results or return for a follow-up appointment to receive results.

Study population, sample and data collection:

- Was the interview conducted at the patient’s healthcare facility, and did they receive any incentive or any subsidy or reimbursement to attend interviews?

- Did any participants require interpreters?

- Who transcribed the interviews and focus group discussions? IDI and FGD were done in Luganda while KII were done in English. Was analysis done in the language of interviews or was translation of transcripts done before analysis? If so who did translation?

- It is unclear whether the data from all three sources (focus groups, patient and participant interviews) were all coded together; and whether both researchers coded all data from all three sources independently.

- Line 181: clarify who participated in these meetings.

Findings:

- Please include a range for the number of participants on each focus group.

- Line 220: ‘three quarters of them were female and an equal number,6, aged between 20-30 years and 31-40 years.’ needs editing for clarity

- The term ‘failure to produce sputum’ comes across as ascribing blame to the individual. Suggest rephrasing to focus on the situation rather than implying fault. (e.g., inability to obtain sputum sample)

- Please make clearer how data analysis combined data from all data sources.

Facilitators for completion:

- Lines 255-256: ‘Obtaining same day results: few participants mentioned receiving results on the same day of being presumed for TB and attributed their completion that.’ – needs revising to improve clarity

- Line 285: ‘Caring health workers motivated patients to get tested for TB’ – it is unclear when you read this sentence whether it is a specific characteristic/behaviour of the healthcare worker that motivated patients. This is however explained later in the paragraph. I suggest the first sentence to be revised – perhaps positive interaction with healthcare workers motivated patients?

- Line 296: cough monitors require definition.

- Line 344: this paragraph seems to talk to barriers and not facilitators. Suggest reviewing and considering the merge with the barriers around ‘contact details’ in page 28.

- Line 355: regarding community facilitators – did healthcare workers speak about these?

Barriers for completion:

- The term ‘failure to produce sputum’ comes across as ascribing blame to the individual. Please consider revising it as suggested above.

- Line 375: ‘they failed to produce sputum when they were given a sputum container to put the sample’ – please revise

- Line 406 ‘presumed for’ is unclear – please modify to include TB

- Line 415: ‘hence stigmatizes those with TB which makes patients fear to be tested’ – edit for clarity

- Line 416: ‘how stigma makes patients not to test for TB’ – consider editing to remove ‘to’ or change to an alternative "e.g., discourages from testing for TB".

- Line 448: percentages aren’t necessary and were not used anywhere else, so I suggest the use of an alternative to convey the significance or frequency of this finding (e.g., more than half).

- Lines 448-456: suggest reviewing this paragraph in full for flow and to improve its readability; also, to improve connection to the rest of the section

- Lines 489-501: suggest reviewing this paragraph in full to improve flow and readability and remove repetition.

- Line 539: ‘made an error’ – please review for clarity.

- Line 540: ‘to extend far when talking to them’ – please review for clarity.

Discussion:

- Lines 571-575: ‘caring health workers’ suggest re-wording for clarity – it does not clearly communicate that ‘caring’ refers to the interaction with a healthcare worker who is caring. Please consider re-wording to communicate a positive interaction with healthcare workers who were helpful, caring, compassionate. ‘Lack of transport’ does not capture the extensive issues including financial issues associated with travel described in earlier sections.

- Line 578: ‘as from’ is incomplete.

- Line 580: ‘and other offer words’ – other is repeated, can be removed.

- Line 592 ‘Further, our formative study showed that over half of those who conducted the TB investigations required, did not receive their test results on the same day of being tested’ – needs re-writing for clarity

- Line 594: is ‘inadequate laboratory personnel’ also a finding deriving from this study? If so ‘in this study’ should be moved – or sentence re-written for clarity.

- Line 596: consider replacing the term ‘positive patients’ as per the Union/ Stop TB guide to use of language.

- Line 601: ‘Participants in this study mentioned that lack transport to return to the health facilities was barrier which further amplified the loss’ – this has already been covered in previous and subsequent sentences.

- Line 610: ‘unclear appointment’ – please add to clarify e.g., unclear appointment times/arrangements.

- Line 651: ‘Although the attitude and behaviour of health workers has been criticized over the years as deterring patients to seek care in health facilities, our study found otherwise’ – Negative attitude has also been described as a finding of this study. This needs to be reviewed.

- Line 660: ‘hence the health care workers show empathy to foster compliance among patients’ I think this diminishes the reasons why healthcare workers may show empathy. Suggest revising it.

- Does the study have any other limitations which should be included? Were there difficulties combining healthcare workers and participants views?

Typographical / grammatical errors:

- Pretreatment and pre-treatment appear throughout – use one for consistency

- Line 25 Methods: ‘in-depth’ instead of in-depths / ‘with patients’ instead of ‘among patients'

- Lines 25-26: could consider using ‘person/people with presumed TB’ and ‘person with TB’ instead of ‘TB patients’ as per language guide. Also consider reviewing the use of ‘case’ in line 22.

- Line 105: night sweets – spelling needs correcting

- Line 176: ‘an over view of the interview guides’ – correct spelling to overview

- Lines 309 and 324: ‘to reduce on’ – remove ‘on’

- Line 318-320: personnel is plural – verb conjugation needs revising (prioritize, ensure)

- ‘Pick results’ appear throughout the manuscript. I suggest changing it to ‘collect results’ or ‘pick up results’

- Line 332: ‘in majority of’ – to be changed to ‘in the majority of’

- Line 336: remove ‘they left behind’

- Lines 287 and 358: ‘in form of’ needs changing to ‘in the form of’ or revise

- Lines 361 and 430: ‘below 35 / 40 years’ – to be revised to ‘under 35 years of age’

- Line 439: ‘hence need’ change to ‘hence the need to’

- Line 443: change to ‘the lack of’ instead of ‘this lack of’

- Lines 462, 524, 618, 631: ‘Moreso’ to be corrected to ‘more so’ or synonym

- Lines 496-497: ‘This was however not always true with some participants mentioning possibilities of not finding results on that day when they are told to return.’ Needs revising for clarity

- Line 499: ‘necessitates the need’- please reword

- Line 535: ‘attitude on’ – suggest changing it to ‘attitude towards’

- Line 632: ‘Literature shows that community often don’t want’ - edit to avoid abbreviation

- Line 659: ‘when tested positive’ can be removed

- Line 682: ‘health-worker’ – edit to health worker without a hyphen.

6. PLOS authors have the option to publish the peer review history of their article (what does this mean? ). If published, this will include your full peer review and any attached files.

**Do you want your identity to be public for this peer review?** For information about this choice, including consent withdrawal, please see our Privacy Policy .

Reviewer #1: **Yes: ** Paddy Mutungi Tukamuhebwa

Reviewer #2: No

---

## [Decision Letter · Decision Letter 1]

10 Jul 2025

PGPH-D-25-00066R1

Facilitators and barriers for completion of the diagnostic process among people with presumed tuberculosis in Central Uganda

Dear Dr. Nuwematsiko,

Thank you for submitting your manuscript to PLOS Global Public Health. After careful consideration, we feel that it has merit but does not fully meet PLOS Global Public Health’s publication criteria as it currently stands. Therefore, we invite you to submit a revised version of the manuscript that addresses the points raised during the review process.

There are only some very minor comments to address, as per reviewer 1, prior to publication. Kindly address these. 

We look forward to receiving your revised manuscript.

Kind regards,

Suman Majumdar

Academic Editor

Journal Requirements:

Reviewers' comments:

Reviewer's Responses to Questions

**Comments to the Author**

1. If the authors have adequately addressed your comments raised in a previous round of review and you feel that this manuscript is now acceptable for publication, you may indicate that here to bypass the “Comments to the Author” section, enter your conflict of interest statement in the “Confidential to Editor” section, and submit your "Accept" recommendation.

Reviewer #1: All comments have been addressed

Reviewer #2: All comments have been addressed

2. Does this manuscript meet PLOS Global Public Health’s publication criteria ? Is the manuscript technically sound, and do the data support the conclusions? The manuscript must describe methodologically and ethically rigorous research with conclusions that are appropriately drawn based on the data presented.

Reviewer #1: Yes

Reviewer #2: Yes

3. Has the statistical analysis been performed appropriately and rigorously?

Reviewer #1: Yes

Reviewer #2: N/A

4. Have the authors made all data underlying the findings in their manuscript fully available (please refer to the Data Availability Statement at the start of the manuscript PDF file)?

Reviewer #1: Yes

Reviewer #2: (No Response)

5. Is the manuscript presented in an intelligible fashion and written in standard English?

Reviewer #1: Yes

Reviewer #2: Yes

6. Review Comments to the Author

Reviewer #1: I confirm that the authors have thoroughly reviewed and addressed the comments raised during the peer review process. I do not have any new comments.

Reviewer #2: Thank you for your thorough revisions and responses to the comments. The changes made to the manuscript have strengthened the overall clarity and presentation of the work. I only have a few remaining minor suggestions on the revised version to further improve consistency, please see below:

1) The term ‘failed’ or ‘failing’ to produce sputum is still used in a few instances, for example lines 423, 424, 426 and 427.

2) Line 58: ‘Pre-diagnosis and pretreatment loss of patients facilitates continued spread of TB in the community, poor treatment outcomes for those infected and death’ – suggest re-wording ‘those infected’ for person/s or those with TB.

3) Line 99 in revised version: please review use of round bracket/parentheses after Mityana District) - there seems to be an extra closing parenthesis at the end.

4) Suggest mentioning that the audios were transcribed by the researchers who conducted the interviews – as per the comment/clarification provided in the review table.

5) The closing sentence of the Conclusion mentions the TB & HIV same day testing policy. I would recommend making it clearer which policy this refers to (ie an existing policy - by study facilities or a ministerial policy etc), and what the stigma is associated with (with the policy, with HIV) so the closing sentence is clear and well defined.

7. PLOS authors have the option to publish the peer review history of their article (what does this mean? ). If published, this will include your full peer review and any attached files.

**Do you want your identity to be public for this peer review?** For information about this choice, including consent withdrawal, please see our Privacy Policy .

Reviewer #1: **Yes: ** Paddy Mutungi Tukamuhebwa

Reviewer #2: **Yes: ** Marcela Lima

---

## [Editor Report · Decision Letter 2]

1 Aug 2025

Facilitators and barriers for completion of the diagnostic process among people with presumed tuberculosis in Central Uganda

PGPH-D-25-00066R2

Dear Nuwematsiko,

We are pleased to inform you that your manuscript 'Facilitators and barriers for completion of the diagnostic process among people with presumed tuberculosis in Central Uganda' has been provisionally accepted for publication in PLOS Global Public Health.

Best regards,

A/Prof Suman Majumdar

Academic Editor